# Contribution of Serum Cytomegalovirus PCR to Diagnosis of Early CMV Primary Infection in Pregnant Women

**DOI:** 10.3390/v14102137

**Published:** 2022-09-28

**Authors:** Claire Périllaud-Dubois, Elise Bouthry, Lina Mouna, Christine Pirin, Corinne Vieux-Combe, Olivier Picone, Anne-Marie Roque-Afonso, Alexandre J. Vivanti, Christelle Vauloup-Fellous

**Affiliations:** 1Laboratoire de Virologie, Hôpital Universitaire Saint-Antoine, Institut Pierre Louis d’Epidémiologie et de Santé Publique, Sorbonne Université, INSERM, AP-HP, 75018 Paris, France; 2Virology Laboratory, AP-HP, Sorbonne Université, Hôpital Saint-Antoine, 75012 Paris, France; 3Virology Laboratory, CHU Angers, 49000 Angers, France; 4Virology Laboratory, AP-HP, Université Paris-Saclay, Hôpital Paul-Brousse, 94804 Villejuif, France; 5INSERM U-1193, Université Paris-Saclay, 94804 Villejuif, France; 6Division of Obstetrics and Gynecology, AP-HP, Nord, Hôpital Louis Mourier, 92700 Colombes, France; 7Division of Obstetrics and Gynecology, AP-HP, Université Paris-Saclay, 92140 Clamart, France

**Keywords:** CMV primary infection, pregnant women, CMV PCR, serum, CMV IgM, valaciclovir

## Abstract

(1) Background: What is the role of serum CMV PCR in the diagnosis of recent primary infection (PI) in pregnant women when IgG avidity is uninformative? (2) Methods: Retrospective cohort study to compare serum versus whole blood CMV PCR. (a) Qualitative assessment: CMV PCR was performed on 123 serum samples and 74 whole blood samples collected from 132 pregnant women with recent CMV PI. PCR positivity rate was used to calculate sensitivity in serum and whole blood. (b) Quantitative assessment: CMV PCR was performed on 72 paired samples of serum and whole blood collected on the same day from 57 patients. (3) Results: In pregnant women, PCR positivity rate was 89% for serum samples versus 100% in whole blood in the case of very recent PI (<15 days), but only 27% in serum versus 68% in whole blood for PI occurring from 6 weeks to 3 months before. Comparing CMV viral loads between serum and whole blood, we determined the limit of CMV DNA detection in serum as 3 log copies/mL (whole blood equivalent). (4) Conclusions: Serum CMV PCR is reliable in confirming PI in cases when only IgM is detected. It is therefore a valuable tool in introducing valaciclovir treatment as early as possible to prevent mother-to-child CMV transmission.

## 1. Introduction

Cytomegalovirus (CMV) is the most frequent cause of congenital viral infection worldwide with an estimated prevalence of between 0.5 and 1% of all live births. Congenital CMV is a major cause of sensorineural hearing loss and mental retardation [1,2,3,4,5,6]. Recently, valaciclovir has proven beneficial in lowering the risk of congenital CMV in first-trimester maternal CMV primary infections (PI), and importantly authors have highlighted that quick initiation of treatment after onset of infection is essential [7,8,9]. Early confirmation of maternal PI is therefore of major importance in management of these patients.

Diagnosis of CMV PI during pregnancy mainly relies on serology: detection of specific CMV IgG and IgM, associated with CMV IgG avidity in the case of positive CMV IgM [10]. Our team recently showed that the positive predictive value of CMV IgM was only 16.4% in systematic screening during pregnancy and confirmed that the CMV IgG avidity test is essential to confirm CMV PI [11]. Moreover, IgG avidity allows estimation of the date of PI and of the risk of fetal infection [12,13]. However, there are two situations where CMV IgG avidity is uninformative: 1) Very early after onset of infection when CMV IgM is positive but CMV IgG is negative or too low to measure avidity, and 2) when the avidity index is moderate and consequently cannot exclude a recent CMV PI less than 3 months before [12]. In these situations, we wondered if PCR in the same serum could be of any help in confirming or excluding a recent PI. Such a recent PI would then prompt the introduction of valaciclovir treatment as soon as possible.

Several studies have already demonstrated greater sensitivity of CMV PCR in whole blood versus plasma [14,15]. In France, whole blood is the recommended matrix for CMV PCR because leucocytes are present in whole blood, and is used to monitor CMV viral load in immunocompromised patients [16]. Unfortunately, whole blood is rarely available in routine management of pregnancy whereas serum samples (liquid as blood plasma without clotting factors) are regularly collected and stored for at least one year in France. Therefore, we aimed to evaluate the performance of CMV PCR in serum to exclude/confirm PI in pregnant women in the case where avidity is not reliable.

## 2. Materials and Methods

### 2.1. Samples

#### 2.1.1. CMV PCR in Serum and Whole Blood Collected from Pregnant Women with Recent PI

Between April 2016 and December 2021, we collected blood samples from 148 pregnant women: 123 serum samples and 74 whole blood samples from 132 pregnant women with known recent PI and 16 serum samples from 16 pregnant women with PI that occurred more than 1 year before (Appendix A). CMV PI was confirmed by seroconversion and/or avidity monitoring in serum. CMV IgM and CMV IgG were measured with Liaison XL (DiaSorin^®^, Saluggia, Italy) and CMV IgG avidity was measured with Vidas (bioMérieux^®^, Craponne, France). Serum and whole blood samples were classified according to the time since onset of infection, according to serological settings [12]:positive CMV IgM with negative CMV IgG: approximately 15 dayspositive CMV IgM, positive CMV IgG and CMV IgG avidity index < 20%: PI between 2 and 6 weekspositive CMV IgM, positive CMV IgG and CMV IgG avidity index between 20 and 40%: PI between 6 weeks and 3 months

Serum samples were stored at −40 °C until CMV PCR, unlike whole blood samples, which were not frozen because PCR was performed on fresh samples.

Concerning the 16 pregnant women with PI that occurred more than 1 year before, we selected 16 serum samples with positive CMV IgM and moderate CMV IgG avidity index (40–65%) for CMV PCR (Appendix A). These were collected from 16 pregnant women known to be seropositive for CMV more than one year before sample collection. Serum samples were previously stored at −40 °C until analysis.

#### 2.1.2. Comparison of CMV PCR in Whole Blood versus Serum

Between March 2018 and October 2021, we chose 72 serum samples from 57 patients (Appendix A) because they were collected in the same patient and on the same day as a whole blood sample with positive CMV PCR. Serum samples were previously stored at −40 °C until analysis. Samples were selected regardless of the clinical situation: PI or non-primary infection (NPI) in immunocompetent or immunocompromised patients (Appendix A).

### 2.2. CMV DNA Detection and Quantification

The same extraction/amplification protocols were used for whole blood and serum samples. For nucleic acid extraction, the automated system QIASymphony (Qiagen, Germantown, MD, USA) with the DSP virus/pathogen minikit was used following the manufacturer’s recommendations. For amplification and detection, the Rotor-Gene Q (Qiagen) with the Artus^®^ CMV QS-RGQ kit (Qiagen) was used, following the manufacturer’s recommendations. CMV DNA was expressed as copy number per milliliter of whole blood or serum.

### 2.3. Statistical Analysis

Statistical analyses were performed with RStudio software (version 1.4.1103). We used non-parametric Wilcoxon tests to compare mean CMV viral loads in whole blood and serum, and to compare mean CMV viral loads in serum after maternal PI. We compared CMV PCR positivity percentages with Chi-squared tests according to the estimated date of onset of PI in pregnant women. Differences were considered statistically different if the *p* value was <0.05.

## 3. Results

### 3.1. Description of Results

#### 3.1.1. Sensitivity and Quantitative Comparison of CMV PCR in Serum versus Whole Blood in Pregnant Women with Recent PI

We performed CMV PCR on the 123 serum samples and 74 whole blood samples collected from 132 pregnant women with known recent PI. We determined PCR sensitivity (as PCR positivity rate) in serum and whole blood, and CMV viral loads in serum and whole blood according to the time lapse between PI and sample collection (Figure 1A,B).

For samples collected approximately 15 days after PI, sensitivity of CMV PCR in serum was 89%, 95CI [64–98%] and sensitivity in whole blood was 100%. Median viral load of positives in serum was 1.9 log copies/mL, IQR [1.6 log–2.5 log] copies/mL and in whole blood was 3.3 log copies/mL [3.1 log–4.0 log] copies/mL (*p* = 0.01). If PI occurred between 2 and 6 weeks before sample collection, sensitivity of CMV PCR in serum was 65%, 95CI [50–78%] and sensitivity in whole blood was 78% 95CI [52–93%]. Median viral load of positives in serum was 1.9 log copies/mL, IQR [1.7 log–2.1 log] copies/mL and in whole blood was 2.9 log copies/mL [2.7 log–3.2 log] copies/mL (*p* = 0.007). In the case of PI occurring between 6 weeks and 3 months, the sensitivity of CMV PCR was 27%, 95CI [17–40%] in serum and 68% 95CI [53–80%] in whole blood (*p* < 0.001). Median viral load of positives was 1.6 log copies/mL, IQR [1.4 log–1.8 log] copies/mL in serum and 2.4 log copies/mL [1.8 log–2.8 log] copies/mL in whole blood (*p* < 0.001). If PI occurred before 6 weeks, quantitative CMV PCR results in serum were significantly higher compared to later (*p* < 0.001) and qualitative CMV PCR in serum was statistically different compared to later (*p* < 0.001).

We performed CMV PCR on the 16 serum samples collected from 16 pregnant women known to be seropositive for CMV more than one year before collection. CMV PCR was negative in all samples (16/16, 100%).

#### 3.1.2. Comparison of CMV PCR in Whole Blood versus Serum

We performed CMV PCR on the 72 serum samples collected from 57 patients on the same day as a positive CMV PCR in whole blood. Serum samples were collected from transplanted or immunocompetent patients with CMV NPI or CMV PI (Appendix A).

Correlation between whole blood and serum CMV viral loads was checked using the 72 samples with a good determination coefficient r² = 0.71 (Figure 2A). Coefficient of correlation was 0.85. Overall, 42/72 serum samples (58%) had a detectable viral load, and paired whole blood viral loads ranged from 55 copies/mL to 6.9 log copies/mL (median 3.2 log copies/mL, IQR [2.9–4.2] log copies/mL). For the other 30/72 serum samples (42%), CMV DNA was not detected, whereas viral loads in whole blood ranged from 4 copies/mL to 3.1 log copies/mL (median 2.0 log copies/mL, IQR [1.4–2.4] log copies/mL) (Figure 2A). Below 3 log copies/mL in whole blood, CMV PCR could be detected in only 16/43 (37%) serum samples. Overall, serum CMV viral load was significantly lower (an estimated 1 log lower) compared to whole blood (*p* < 0.001) (Figure 2B). We determined a limit of detection (LOD) of 3 log copies/mL (whole blood equivalent) for serum CMV viral load.

## 4. Discussion

To diagnose CMV PI in pregnant women, serology is agreed to be quite reliable, and is widely used. However, sometimes CMV IgG avidity may be uninformative and retesting on a serum sample collected at least one week later may be requested to confirm PI. In our retrospective cohort study, we evaluated the input of CMV PCR in serum in confirming/excluding PI in pregnant women compared to CMV PCR in whole blood and depending on the time since onset of PI (from 15 days to 3 months). CMV PCR sensitivity in serum and whole blood was respectively 89% and 100% if PI occurred approximately 15 days before, 65% and 78% if PI occurred 2 to 6 weeks before, and 27% and 68% if PI occurred 6 to 13 weeks before (*p* < 0.001). In very recent PI (approximately 15 days), CMV PCR in serum is quite reliable in confirming PI, as it is positive in 89% of cases. Moreover, before six weeks post-PI, PCR positivity rates were not statistically different between serum and whole blood. Later (six weeks to three months after PI), CMV PCR in serum is informative only if positive, as it is positive in only 27% of cases.

The main interpretations of our findings are the following:after CMV infection, CMV IgM is positive approximately 18 days later, which is concomitant with a positive CMV PCR both in whole blood and serum.IgG seroconversion appears at Day 20 and IgG avidity is reliable from Day 25/30.

Therefore, by performing CMV PCR in serum rather than waiting for the IgG avidity index, 7 to 12 days can be gained in introducing valaciclovir.

We compared CMV viral loads in both serum and whole blood samples collected on the same day in order to determine whether CMV PCR results are reliable in serum. As expected, CMV viral loads in serum were significantly lower compared to whole blood (loss of sensitivity in serum compared to whole blood). Focusing on high viral loads (over 3 log copies/mL in whole blood), viral load in serum correlated well with whole blood. However, below 3 log copies/mL in whole blood, detection in serum was not reliable: viral load in serum was detected in only 37% of cases. As we observed a significant decrease of CMV viral load after six weeks post-PI, this could explain why serum PCR results are mostly negative six weeks to three months after PI. One limitation of our results is that CMV PCR in serum was not performed on fresh samples and that the freeze-thaw cycle may have had a negative impact on the PCR result. Serum samples were stored at −40 °C, whereas −80 °C would probably have been more appropriate. On the other hand, CMV PCR in whole blood was performed on fresh samples. Moreover, if CMV PCR was performed on serum samples prospectively, it would be performed on fresh serum samples, which could increase performance compared to our observations.

Lower sensitivity in serum has been reported in several studies. Andrade et al. [17] and Chen et al. [18] reported that the sensitivity of PCR in serum was lower than in whole blood, but with a higher PPV in predicting CMV infection in immunocompromised patients: PPV of 70% in serum versus 52% in whole blood. Our results show that PCR in serum is 1 log lower compared to whole blood, which is concordant with results reporting comparison between plasma and whole blood [19], but should be confirmed in a prospective study to avoid freeze-thaw cycles of serum samples.

There are very few available data on CMV PCR kinetics in serum collected from pregnant women with CMV PI. Berth et al. [20] reported that most immunocompetent patients had negative PCR in whole blood four weeks after CMV PI, but Revello et al. [21] showed that CMV PCR is likely to be positive in whole blood within one or two months after PI in pregnant women. Here, we estimated that CMV PCR in serum is no longer reliable more than one month after PI in pregnant women and nor can we consider using CMV PCR in serum to exclude PI in the case of moderate IgG avidity index. Globally, a positive PCR in serum allows us to affirm recent PI whereas a negative PCR cannot rule out the diagnosis. Although CMV PCR in serum has limited sensitivity in diagnosis of CMV PI, it could really help in the case of isolated CMV IgM. Currently, if CMV IgM is positive but CMV IgG is negative, we recommend a subsequent serologic test on a serum sample collected a week later in order to distinguish non-specific IgM from PI. When PI is confirmed, treatment with valaciclovir lowers the risk of fetal infection, as recently described [7,8]. Therefore, in this situation where IgG avidity cannot help (because IgG is negative or too low), CMV PCR on serum is a valuable tool that enables valaciclovir to be initiated as early as possible. Nevertheless, as we assume that PCR in whole blood is more sensitive than PCR in serum, in the case of isolated IgM and negative PCR in serum, it would be wise to perform CMV PCR on a whole blood sample collected as soon as possible. Our study focuses on the importance of early CMV PI diagnosis in order to treat to prevent CMV infection, but we obviously highlight the crucial importance of avoiding and preventing the maternal infection, which requires knowledge in pregnant women [22]. 

Our study encourages the use of CMV PCR on serum samples with positive CMV IgM and negative or low titers of CMV IgG in order to introduce valaciclovir as early as possible to prevent mother-to-child CMV transmission. If we discuss the current use of valaciclovir in pregnancy, it is not yet fully integrated in routine. Another therapeutic option to prevent mother to child transmission could be hyperimmunoglobulin (HIG) [23]. In the same way, CMV PCR on serum would be useful to introduce HIG as soon as possible. PCR in serum can also be performed later, in a few weeks after PI, but only a positive result will be informative in this case. Further studies will be useful in evaluating CMV IgG avidity and CMV PCR (both in serum and whole blood) in women treated with valaciclovir, in order to investigate whether very early treatment has an influence on IgG kinetics, avidity maturation and/or persistence of detectable CMV DNA.

## Figures and Tables

**Figure 1 viruses-14-02137-f001:**
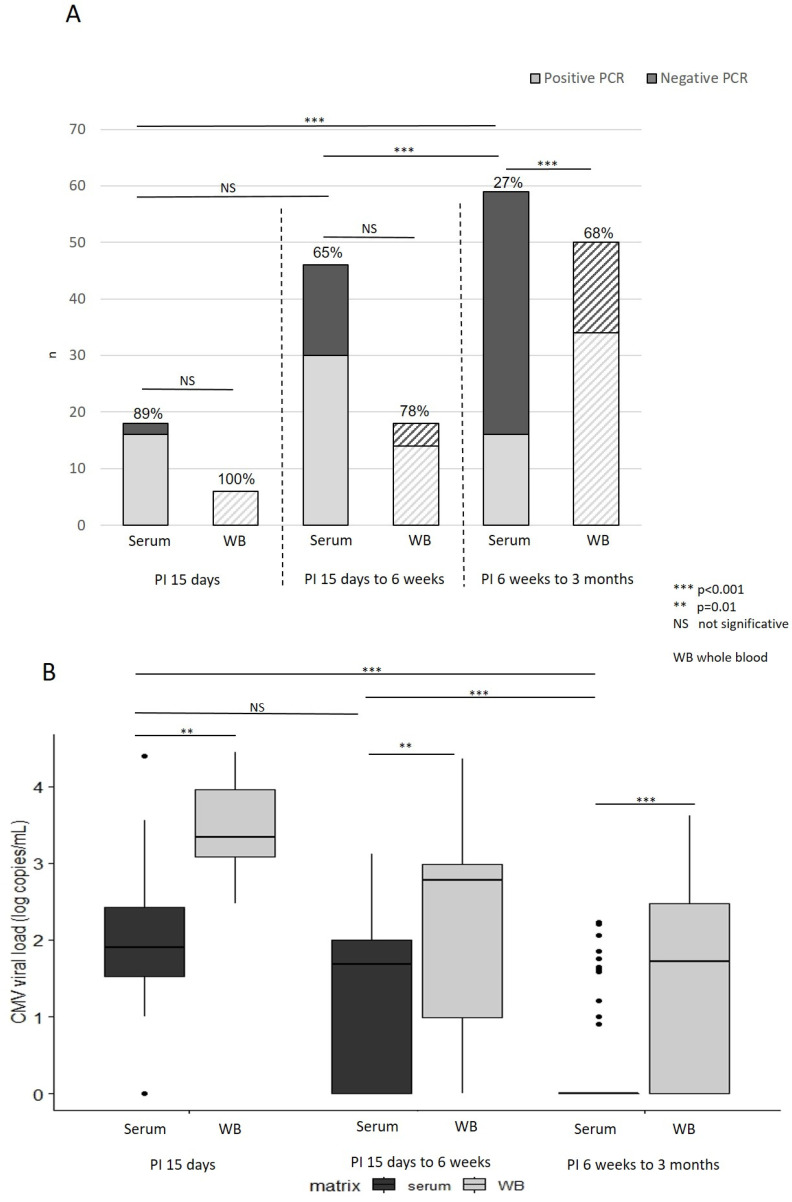
Qualitative and quantitative performances of CMV PCR in serum and whole blood in pregnant women with recent PI. (**A**) Sensitivity of CMV PCR in serum and in whole blood according to the delay after primary infection (PI) in pregnant women. Barplot representing percentage of serum or whole blood samples with positive CMV PCR (sensitivity) depending on the time lapse since the onset of PI: Light grey bars represent negative CMV PCR and dark grey bars represent positive CMV PCR. Serum samples are colored and whole blood samples are represented with hatched lines. Chi-squared test results are given so as to compare PCR sensitivity according to the matrix (serum versus whole blood) and according to the time lapse since PI. (**B**) Quantitative CMV PCR results in serum and in whole blood according to the time interval since (PI) in pregnant women. Boxplot representing CMV viral loads in serum and whole blood depending on the delay time since the onset of PI: Grey plots represent serum samples and yellow plots represent whole blood samples. Wilcoxon test results are given so as to compare CMV viral loads according to the matrix (serum versus whole blood) and according to the time lapse since PI. PI: primary infection. ***: *p* < 0.001; **: *p* = 0.01; NS: not significant.

**Figure 2 viruses-14-02137-f002:**
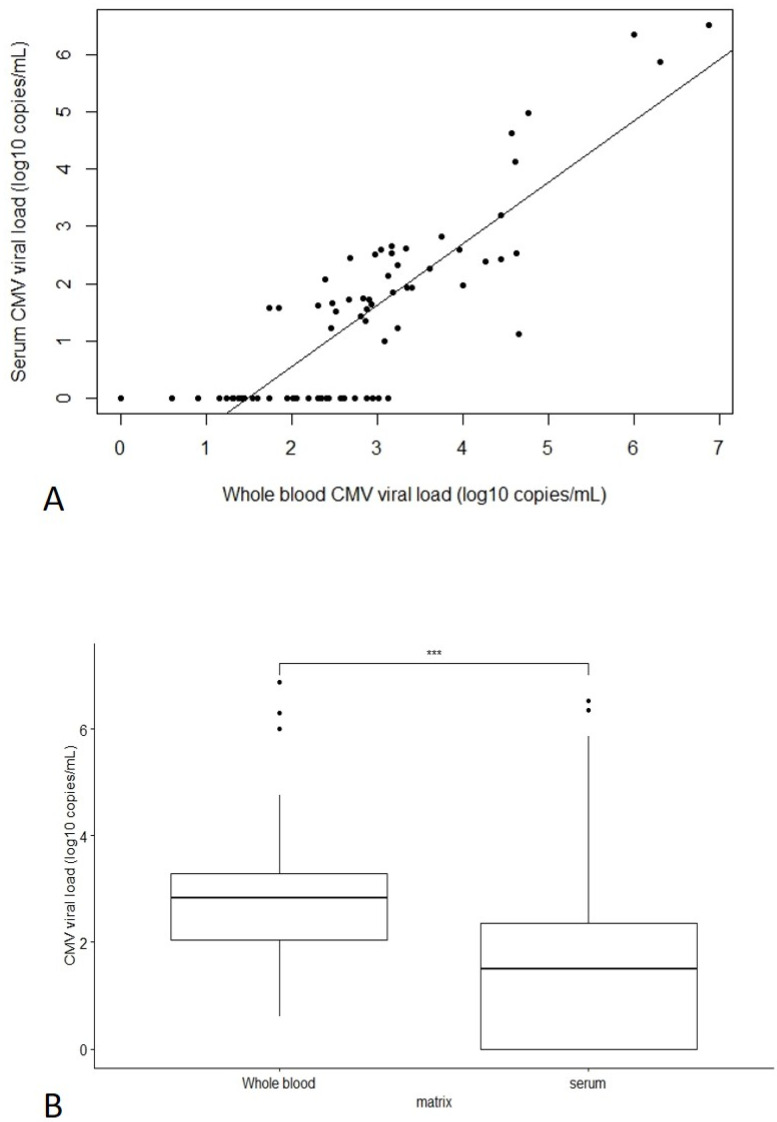
Comparison of Quantitative CMV PCR in Whole Blood versus Serum. (**A**) Plot of correlation between CMV viral load in serum versus whole blood, with linear regression line. PCR undetectable in serum was plotted at 0. Estimated R² was 0.71. (**B**) Boxplot for comparison between CMV viral load in whole blood and serum. Significant difference of means (***) observed using the Wilcoxon test (*p* < 0.001).

## Data Availability

Data can be provided by the authors on demand.

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
