# Peer review of "Contribution of Serum Cytomegalovirus PCR to Diagnosis of Early CMV Primary Infection in Pregnant Women"

_viruses, 2022, doi:10.3390/v14102137_

Round 1

Reviewer 1 Report

This study is interesting and important to the field.

I would like to see an improvement in the English used, as some sentences require a second reading in order to be understood. (e.g. the last setence of the Abstract). There are also several spelling mistakes.

The labelling on the figures is not always easy to read, particularly fig. S2.

It should be made clear that while the use of Valaciclovir in pregnancy shows promise, it is not in routine use or completely effective. The potential use of HIG should also be mentioned in this context.

Otherwise, the paper is well put together and clear.

Author Response

We thank the reviewer for his carrefully reading and reviewing of the manuscript.

Point 1: I would like to see an improvement in the English used, as some sentences require a second reading in order to be understood. (e.g. the last setence of the Abstract). There are also several spelling mistakes.

Response point 1: We extensively improved the English throughout the manuscript with the help of a native english speaker.

Point 2: The labelling on the figures is not always easy to read, particularly fig. S2.

Response point 2: We have redesigned the figure S2 and the labels to make it clearer.

Point 3: It should be made clear that while the use of Valaciclovir in pregnancy shows promise, it is not in routine use or completely effective. The potential use of HIG should also be mentioned in this context.

Response point 3: We mentioned this point in the discussion as suggested.

Lines 249-252: "If we discuss the current use of valaciclovir in pregnancy, it is not yet fully integrated in routine. Another therapeutic option to prevent mother to child transmission could be hyperimmunoglobulin (HIG) [22]. In the same way, CMV PCR on serum would be usefull to introduce HIG as soon as possible. "

Reviewer 2 Report

Dear authors,

I've appreciated your paper, the topic is of great interest and the research well presented

I would like to give minor suggestions

1) please be more clear in the difference between whole blood and serum, people not that much involved in laboratory may benefit for a more clear definition

2) pay attention to the spaces between each word

3) use a better formula of the sentence when talking about IGG close to IGM to do not confuse them in the interpretation of the sentence

4) I would suggest to add few sentence in the discussion, related to the crucial importante of letting women become aware of the existence of this congenital infection, moreover to train them in how to avoid the contact, with this purpose I suggest to read and cite the following paper PMCID: PMC5554994

after these minor revisions I would recommend it for publication

Author Response

We thank the reviewer for his carreful reading and reviewing of our manuscript.

1) please be more clear in the difference between whole blood and serum, people not that much involved in laboratory may benefit for a more clear definition

Response 1: We added a few words to define whole blood and serum in Introduction.

Line 53 and line 55/56

2) pay attention to the spaces between each word

Response 2: we checked all the manuscript for this particular point.

3) use a better formula of the sentence when talking about IGG close to IGM to do not confuse them in the interpretation of the sentence

Response 3: we reformuled sentences to be clearer about IgG and IgM :

4) I would suggest to add few sentence in the discussion, related to the crucial importante of letting women become aware of the existence of this congenital infection, moreover to train them in how to avoid the contact, with this purpose I suggest to read and cite the following paper PMCID: PMC5554994

Response 4: We agree with you about the importance of measures of prevention. We added a sentence in the discussion:

Lines 243-246: "Our study focuses on the importance of early CMV PI diagnosis in order to treat to prevent cCMV infection, but we obviously highlight the crucial importance of avoiding and preventing the maternal infection, which requires knowledge in pregnant women [22]."
